# Online Planning for Stochastic Collaborative Privacy Preserving Planning

## Abstract

Collaborative multi-agent privacy preserving planning (CPPP) models problems where agents must work together to achieve joint goals, while keeping some information private. Recently, CPPP was extended to the stochastic case, where actions may fail, producing different effects than intended. Stochastic CPPP (SCPPP) problems can be solved using offline algorithms, such as RTDP. However, in many cases, we are not interested in computing a complete policy offline, and prefer to use an online approach, where one decides online on the next action only, without exploring the complete state space. This can allow us to scale to much larger problems. In this paper we thus explore online approaches for SCPPP. We suggest using a variant of the well known FF-Replan approach, adapted to CPPP, and a plan repair approach, where we try to locally return to the plan if an undesirable effect has occurred. We provide an empirical evaluation, comparing our approaches to an offline solver, showing that we can scale to much larger problems, and analyzing the strengths and weaknesses of our methods.

## Introduction

In many real world applications several autonomous agents need to collaborate to achieve collective goals. In some cases, these agents are constrained to keep certain information private, not disclosing it to the other agents they collaborate with. *Collaborative Privacy-Preserving Planning (*CPPP*)* models such multi-agent planning tasks, where agents need to collaborate without revealing private information (Brafman and Domshlak 2008). Each agent has private facts and actions that cannot be shared with the other agents. CPPP has important motivating examples, such as planning for organizations that outsource some tasks.

While classical CPPP considers deterministic domains, there are many real world scenarios that are naturally stochastic. That is, when an action is executed, different effects may occur, and one can define a distribution over the possible effects of the action. For example, a robotic arm that organizes objects in a specific structure may wrongly drop the object it is holding, may accidentally hit other objects and move them to other locations, and may successfully put the object in its designed location. One can often know in advance the probability of each of these outcomes. *Stochastic Collaborative Privacy-Preserving Planning (*SCPPP*)* models CPPP problems where actions may have different effects

with varying probabilities (Hefner, Shani, and Stern 2022). Stochastic planning domains can be modeled using *Markov Decision Processes (MDPs)* (Kolobov 2012), and SCPPP is also designed as an MDP.

Hefner, Shani, and Stern (2022) show that SCPPP problems can be solved using MDP based algorithms, such as RTDP. RTDP is a popular approach for solving a goal-based MDP that operates by iteratively executing trajectories in the state space (Barto, Bradtke, and Singh 1995). Hefner, Shani, and Stern (2022) adopt RTDP algorithm for solving SCPPP problems, suggesting two variations: DRTDP and PS-RTDP, varying in the messages sent between agents.

RTDP is an offline approach, where a policy, a mapping from state to actions, is computed before the agent begins to act. During acting, however, in many cases the agent will only visit a relatively small number of states in each execution. In these cases, when the number of executions is low, computing a policy for many states that will not be visited can be wasteful. Online approaches, on the other hand, do not compute a policy before starting to act. Instead, while acting, online approaches make local decisions at each state that is visited (Hansen and Zilberstein 2001). These approaches require little time before beginning to act, but may require significant computations before each action is taken.

In this paper we propose an online planning approach for SCPPP. The agents make a decision about the next action to execute, using a heuristic method (Pearl 1984). We suggest 3 different heuristic strategies for deciding on the next action. First, we suggest computing a privacy preserving cost-to-go estimation (Štolba and Komenda 2014), extended to the non-deterministic case. Then, we follow the well-known FF-Replan approach (Yoon, Fern, and Givan 2007), where a classical plan is computed to decide on the next action. We compute a plan using a classical CPPP approach (Nissim and Brafman 2014). That plan is followed until an unexpected action outcome occurs, at which point we replan. Finally, we suggest a plan repair approach, augmenting our FF-Replan method. When an unexpected outcome occurs, instead of replanning for all agents, the agent that experienced the unexpected outcome attempts to replan locally to return to the existing joint plan. Only if the agent fails, then a complete replan is launched.

We provide a wide set of experiments over benchmarks based on the CODMAP competition (Štolba, Komenda, and

Kovacs 2015). We construct stochastic problems with richer stochastic effects. We first shows that the offline RTDP approach only solves very small problems while we scale to many orders of magnitude larger instances. We then compare our online methods, both in terms of policy quality, as well as in terms of the required time for each decision. We show that the cost-to-go heuristic is often much faster to compute, but sometimes makes low quality decisions. In domains where actions can fail in several ways, our plan repair method can decide faster on the next action.

## Background

A multi-agent problem (Brafman and Domshlak 2013) is a tuple $\langle P, \{A_i\}_{i=1}^{k}, I, G \rangle$ where: $k$ is the number of agents, $P$ is a finite set of facts, $A_i$ is the set of actions agent $i$ can perform, $I$ is the start state, and $G$ is the goal condition.

Each action $a = \langle pre(a), eff(a) \rangle$ is defined by its preconditions ($pre(a)$), and effects ($eff(a)$). Preconditions and effects are conjunctions of facts, or fact negations. A state is a truth assignment over $P$, typically modeled by the set of facts that hold in $s$. $G$ is a conjunction of facts. $a(s)$ denotes the result of applying action $a$ to state $s$. A *plan* $\pi = (a_1, \ldots, a_k)$ is a solution to a planning task iff $G \subseteq a_k(\ldots (a_1(I) \ldots)$.

Collaborative privacy-preserving planning is a multi-agent problem where for each agent $i$ there is a disjoint subset of facts and a disjoint subset of actions that are private, known only to agent $i$, denoted $private_i(P)$ and $private_i(A_i)$, respectively. $public(P)$ is the subset of public facts in $P$ that are known to all agents. $public_i(A_i)$ is a set of public actions of agent $i$. As opposed to private actions, the execution of a public action is observed by all agents, and the public effects that it generates. Some preconditions and effects of a public action may be private, and the action obtained by removing these private elements is called its *public projection*, and it is known to all agents. Goals can be public or private to an agent. The problem is collaborative, that is, all agents work together towards achieving both the public and the private components in $G$.

An agent is aware only of its *local view* of the problem, that is, its private actions and facts, its public actions, the public facts, and the public projection of the actions of all other agents. For public actions of other agents, the local view contains only the public preconditions and effects. During execution, we assume that all agents observe the execution of a public action, and the changes in the public facts.

MAFS is a distributed algorithm which computes a complete plan using collaborative forward search (Nissim and Brafman 2014). Each agent maintains an open list of states, and in every iteration each agent chooses a state in the open list to expand, generating all its children and adding them to the open list (avoiding duplicates). Whenever an agent $i$ expands a state that was generated by applying a public action, it also broadcasts this state to all other agents. An agent $j$ that receives a state adds it to his open list.

To preserve privacy, the private part of a state is obfuscated, e.g., by replacing the private facts with some index, such that only the broadcasting agent knows how to map this index to the corresponding private facts. Also, to avoid some privacy leakage, one can send new indexes for an already sent state (Brafman 2015). Once the goal is reached, the agent achieving the goal informs all others, and the search process stops.

Stochastic CPPP is an extension of CPPP to stochastic domains, modeled as an MDP, with each agent viewing only a part of the complete MDP (Hefner, Shani, and Stern 2022). More formally, a Stochastic Privacy Preserving Planning problem (SCPPP) is a tuple $\langle P, \{A_i\}_{i=1}^{k}, \{tr_i\}_{i=1}^{k}, I, G, C \rangle$ where $P$, $k$, $I$, and $G$ are as above. $A_i$ is the set of stochastic actions for agent $i$. $\{tr_i\}_{i=1}^{k}$ is a transition function for agent $i$ specifying its probability of moving between states by performing actions. $C$ is a cost function.

As in a CPPP problem, $private_i(P)$ and $private_i(A_i)$ denote private facts and actions of agent $i$. $public(P)$ is the set of public facts in $P$, and $public_i(A_i)$ is the set of public actions of agent $i$. As in an MDP, actions have stochastic effects, i.e. $eff(a) = \{\langle \phi_i, p_i \rangle\}$ where $p_i$ is the probability that effect $\phi_i$ would occur and $\sum_i p_i = 1$ and $C$ assigns a cost for executing an action at a state. In this paper, we only allow non-negative costs. Agent $i$ knows only the transition function for actions in $A_i$.

While in classical CPPP problems a solution is a sequence of public and private actions, in a SCPPP problem the solution is a policy, determining for each state which agent should execute an action and which action must be executed. However, each agent only sees the public part of the state, and its own private state. To preserve privacy, many algorithms model the global state by a tuple $\langle s_{pub}, \tilde{s}_1, ..., \tilde{s}_k \rangle$, where $s_{pub}$ the set of public facts that hold, and for each agent $i$, $\tilde{s}_i$ is an index that $i$ maps internally to $s_i$, the set of private facts of $i$ that hold. Using this representation we can maintain a two-level policy. At the higher level we maintain a mapping from $\langle s_{pub}, \tilde{s}_1, ..., \tilde{s}_k \rangle$ to agents, denoting which agent should act in the current global state. Then, each agent maintains a local private policy specifying which action to execute at the current global state. After the action is executed, the agent $i$ that executed the action sends a new global state, with revised $s_{pub}$ and a new $\tilde{s}_i$.

In this paper, we focus on policies that do not allow parallel execution. We leave extensions that allow for parallel executions where more than one agent executes an action at each step to future research.

In offline planning, the agent computes a solution, i.e., a policy, before starting to act. Then, during policy execution, the agent only queries the policy on which action to execute next. Offline planning algorithms must first compute a policy for every state that might be visited during the policy execution. These algorithms require a significant computational effort before acting, but little effort during policy execution.

Real-Time Dynamic Programming (RTDP) is such an offline MDP algorithm (Barto, Bradtke, and Singh 1995; Bonet and Geffner 2003) which operates by running simulated trajectories in state space. A trajectory begins at the initial state and is advanced until a goal is reached. To advance a trajectory, a heuristically best action is selected and a Bellman update is performed. After the update, a new state $s'$ is selected from the $tr(s, a, \cdot)$ distribution and the trajectory is advanced to $s'$. RTDP is anytime, and can be stopped

before the value function converges, and often obtain a good policy. DRTDP is a direct adaptation of RTDP to SCPPP.

Public Synchronization RTDP (PS-RTDP), the approximate version of DRTDP, relies on the intuition that agents need to collaborate mostly after performing public actions, as opposed to private actions which do not directly affect other agents. As in MAFS, following a private action of agent $i$, the next action should also be of agent $i$. Hence, in PS-RTDP, an agent chosen to execute the next action continues to execute additional actions until it executes a public action. Then, the agents select the best agent to advance the trajectory and progresses it farther.

Online planning algorithms, on the other hand, avoid computing a complete policy before acting. Instead, online algorithms perform planning before deciding on the next action to execute. These algorithms hence trade the costly complete policy computation prior to acting, with local computations prior to every action execution.

Typically, offline algorithms that consider the entire policy can make optimal decisions, while online planning algorithms that must make only rapid local computations, often do not guarantee optimality. In some cases, there is a tradeoff between the amount of time spent deciding on the next action, and the optimality of the decision (Kocsis and Szepesvári 2006). On the other hand, online algorithms that can consider only states that were visited during a particular execution, can scale to much larger problems.

A well known online method for single agent MDP is the FF-REPLAN algorithm (Yoon, Fern, and Givan 2007). FF-REPLAN creates a deterministic variation of the problem and produces a plan for that problem by using a deterministic planner (Hoffmann and Nebel 2001). Then, the agent acts according to the produced plan until it either reaches the goal, or an unexpected state, due to a stochastic effect. When reaching an unexpected state, it re-plans by solving a new determinization of the problem where the unexpected state is the initial state. Our methods are built on an adaptation of FF-Replan to a privacy preserving multi agent scenario.

## Related Work

Several online approaches for MDPs where suggested in the past. Hansen and Zilberstein (2001) suggest the $LAO^*$ algorithm, that uses local heuristic search online to decide on the next action. In their experiments they use a cost-to-go heuristic estimation. RTDP can also be used as an online algorithm, running a number of trajectories before every action decision. The UCT algorithm (Kocsis and Szepesvári 2006) is a model free approach, that explores a partial forward plan tree, using forward simulations to produce heuristic estimates at the leaves of the tree. UCT variants were shown to be successful in probabilistic planning problems (Keller and Eyerich 2012). Online versions of RTDP and UCT were shown to have very similar performance in MDPs (Kolobov, Weld et al. 2012). Algorithms based on Monte-Carlo tree search were also suggested for online planning in MDP (Feldman and Domshlak 2014), also in the multi-agent case without privacy concerns (Choudhury et al. 2022). Adapting MCTS, UCT, and online RTDP to SCPPP is interesting to explore.

There are two main approaches for solving privacy preserving planning problems. The first approach computes a public plan first, containing only public actions. Then, each agent attempts to extend its part in the public plan using private actions. If an agent fails, a new public plan must be computed. For example, the GPPP algorithm (Maliah, Shani, and Stern 2018a) computes a rich public projection of the problem, and computes the public plan in a central manner over this projection. In the second approach all agents plan jointly, informing other agents when they achieve some public state. The most popular representative of this approach is the MAFS algorithm (Nissim and Brafman 2014), which motivated the PS-RTDP algorithm (Hefner, Shani, and Stern 2022) for solving stochastic CPPP problems.

We can incorporate methods based on public plans to SCPPP as well, creating a public policy first, and then local private policies for each agent. Offline-online hybrid, where a public policy is constructed offline, but agents make online choices on their private actions, may be useful.

Many heuristics developed originally for classical planning were adapted to CPPP (Štolba and Komenda 2014; Stolba, Fiser, and Komenda 2015, 2016; Stolba and Komenda 2017; Stolba et al. 2019; Maliah, Shani, and Stern 2014, 2017, 2018b). It is interesting to investigate the use of these heuristics for the stochastic case. Most heuristics would require some adaptation to handle stochastic effects in an informed manner.

As many other CPPP algorithms, we publish heuristic estimates, which may affect privacy. The CPPP community has yet to establish a formal model for privacy which both outlines what must be kept hidden, and allows for efficient computations (Tozicka, Stolba, and Komenda 2017; Stolba, Urbanovská, and Komenda 2022). There is also work in the planning community on learning action models from observing policy executions (Yang, Wu, and Jiang 2007; Zhuo et al. 2010; Juba and Stern 2022). An investigation of these issues is important, but outside the scope of this paper.

## Heuristic Online SCPPP

We now introduce our online algorithm for solving SCPPP problems. At each step, the algorithm chooses which agent should act. Then the agent decides which action to execute. These decisions are based on a heuristic estimate that the agents compute jointly, in a privacy preserving manner. When we discuss heuristics, we sometimes mean a heuristic function, which assigns a value for a state or action, often the cost-to-go until reaching the goals. In other cases, however, we consider heuristic in a broader sense, as a rapid method for making a decision, that is not guaranteed to be correct or optimal (Pearl 1984; Romanycia and Pelletier 1985).

Before executing an action, the algorithm requires two steps. First, the agents must decide on the next agent to act, based on the particular heuristic method. Then, the agent whose action is preferred, takes the lead. That agent executes its seemingly best action, sends the new state to all agents, and then the process is repeated. The execution is terminated when all agents agree that the goal has been reached.

We suggest three different heuristics for choosing actions.

First, we use a simple classical heuristic, $h_{ff}$. Second, we follow FF-REPLAN, solving a determinized CPPP version of the SCPPP problem, and follow its solution until an unexpected state has been reached. Finally, we suggest a plan-repair approach, where when an agent that executed an action reaches an unexpected state, only that agent replans locally to achieve its next public action.

To support privacy preserving, we use a message passing mechanism similar to that in PS-RTDP (Hefner, Shani, and Stern 2022) where each message contains a public state, indexes of private states and information about heuristic estimations for computing heuristics collaboratively.

## Online Algorithm

Algorithm 1 describes our online SCPPP algorithm. Each agent continuously executes it until all agents report reaching the goal (line 2). Once agent $i$ achieves its goal it notifies all other agents (line 4).

In the main loop, each agent processes its received messages (line 5). A message $m$ is a tuple $\langle s, t, i \rangle$ where $s$ is a state, $t$ is the message type and $i$ is the sending agent. There are two types of messages: *goal* and *trajectory* messages. *Goal* message indicates that agent $i$ has achieved its private goal, and all public goals are satisfied. The receiving agents record this information (line 11).

*Trajectory* messages inform that agent $j$ executed an action and observed a new state $s$. That agent sends a new obfuscated identifier for its new private state, and all agents update their current state (line 13). Following the action, the agents must again choose which agent should act next. This decision is specific to the heuristic approaches that we suggest, and is hence left unspecified in Algorithm 1.

After processing its received messages, the *chosen* agent executes its seemingly best action $a^*$, observes the new private state $s_c^{i'}$, and sends a trajectory message to inform other agents (lines 16-18).

## Heuristics

We use a heuristic method to choose the best agent and action to perform at given state $s$. We now review 3 different methods. First, we use a heuristic value estimation based on the well known $h_{ff}$ (Hoffmann and Nebel 2001). We then implement an FF-REPLAN approach, where the agents employ a privacy preserving classical planning algorithm to create a plan for a determinization of the domain. Finally, we augment this approach with plan repair, in cases where an unexpected state was reached.

**Multi-Agent Heuristic Cost-To-Go Estimate**   Our first approach computes an heuristic estimate of the cost-to-go, using a heuristic method developed for classical CPPP planning (Štolba and Komenda 2014). Many such heuristics were investigated for CPPP, but in this paper, we use a distributed computation of $h_{ff}$, which we explain below.

The $h_{ff}$ heuristic constructs a delete relaxation state space by iteratively applying all possible actions at by all agents. The state is constructed in a distributed manner, where the public part of the state is shared, and each agent maintains its private part of the state separately. In our

---

**Algorithm 1:** Online-SCPPP for agent $i$

1  **online-planner**($i$)
2     **while** $\exists j : j$ *did not report goal* **do**
3         **if** $s_c^i$ *is goal state for* $i$ **then**
4             broadcast $\langle s_c^i, goal, i \rangle$
5         **process-messages**()
6         **if** $chosen = i$ **then**
7             **advance-trajectory**()
8  **process-messages**()
9     **foreach** *Message* $m = \langle s, t, j \rangle$ **do**
10         **if** $m.t$ *is goal message* **then**
11             Record that $j$ reports goal
12         **if** $m.t = trajectory$ **then**
13             $s_c^i \leftarrow m.s$
14             $chosen \leftarrow$ agents choose the next agent to act
15  **advance-trajectory**()
16     $a \leftarrow$ **best-action**($s_c^i$)
17     Execute $a$, observe new state $s_c^{i'}$
18     send $\langle s_c^{i'}, trajectory, i \rangle$

---

stochastic case, if an action has several possible effects, all possible effects are added to the next level. This process is continued until no new effects can be obtained. As opposed to the regular $h_{ff}$, in stochastic environment a fact $p$ can be achieved multiple times in different layers of the constructed graph and it is not guaranteed that the cost of achieving $p$ in the earlier layers will be better than the later. After the fact graph was developed, the agents jointly compute a plan in this relaxed space, and the cost of the relaxed plan is the heuristic estimate.

In our case, we must decide on an agent to act. Hence, each agent simulates the execution all its applicable actions. Then, from each resulting state, we run an $h_{ff}$ computation. That is, the agents run jointly numerous $h_{ff}$ computations, and then we chose the action that provided the best heuristic cost-to-go. In particular, for this heuristic method, agents publish their best heuristic cost-to-go estimates, and the agent with the best estimate is chosen to act. That agent chooses the action that produced the best heuristic estimate.

**Deterministic Replanning**   Our second approach does not rely on a heuristic cost-to-go estimation. Instead, we directly compute the next action to execute, following the FF-REPLAN approach for single agent MDP. This is done by creating a deterministic version of the multi-agent CPPP problem, where the most probable effect is the only possible effect. This relies on the underlying assumption that applies in many planning domains, that successful execution of an action is more likely than failure. If this assumption does not hold, one may use a different determinization, where the planner can choose which of the possible effects occur at every action execution.

The determinized problem is created in a distributed manner, where every agent determinizes its own actions, and publishes the public view of its determinized public actions.

| Domain | Cost | | | | | | Total time (sec) | | | | | |
| | Offline | | | Online | | | Offline | | | Online | | |
| | DRTDP | RTDP-CTG | RTDP-DP | CTG | DR | DPR | DRTDP | RTDP-CTG | RTDP-DP | CTG | DR | DPR |
|---|---|---|---|---|---|---|---|---|---|---|---|---|
| CBLE-1 | 4.01 | 4.48 | 4.06 | **3.4** | 4.2 | **3.4** | 0.04 | **0.03** | 2.14 | 0.07 | 2.53 | 1.93 |
| CBLE-2 | 28.98 | 6.62 | 6.53 | 6.2 | 6.8 | **5.6** | 37.84 | 0.55 | 50.15 | **0.09** | 4.58 | 5.75 |
| CBLE-3 | - | **8.96** | - | 13 | 10 | 10 | - | 68.51 | - | **1.45** | 5.77 | 9.08 |
| CBLE-4 | - | - | - | 14.8 | **12.8** | 13.6 | - | - | - | 9.91 | **7.91** | 10.62 |
| DLE-2 | 7.3 | - | 7.22 | **2.2** | 6.8 | 6.8 | 3.06 | - | 7.36 | **0.06** | 2.54 | 2.57 |
| DLE-3 | - | - | - | 28 | **20.8** | 21.6 | - | - | - | **0.56** | 4.91 | 6.85 |
| LG-7 | 26.96 | 26.85 | 26.93 | **26.75** | 27.15 | 28.4 | 183.73 | 40.6 | 181.84 | **0.18** | 3.03 | 3 |
| LG-19 | 21.68 | **21.66** | **21.66** | 31.8 | 23.95 | 23.3 | 166.22 | 12.54 | 162.84 | **0.41** | 2.66 | 2.84 |

Table 1: Comparing cost (number of actions) and runtime (execution time in seconds) for both offline and online algorithms. Best algorithm results are in bold.

Creating the deterministic domain description can be done once, and then in each replanning episode we only need to set the current start state. Then, we can run any off-the-shelf CPPP solver to obtain a plan. This plan is created in a distributed manner, where each agent maintains its own portion of the plan. The agent that is the first to act becomes the *chosen* agent.

Computing a deterministic plan is a costly operation, and we prefer to use the plan as long as it is applicable. That is, we execute the plan until an unexpected stochastic effect occurs, which typically amounts to an action failure. To identify such failures, we modify the CPPP planner such that the agents maintain, in addition to the plan, the sequence of states that should occur along the plan. Then, we can check following an action execution if the resulting state is the same as the expected state. If the state is as expected, we continue to use the current deterministic plan. If it is not, then we replan from the current state.

**Plan Repair** Finally, we suggest a method that allows us to avoid complete replanning, attempting to return to the already computed deterministic plan. To achieve that, we leverage the structure of an CPPP plan. Such plans are built by a sequence of public actions, sometimes called the plan schema. The public schema is interleaved with private agent actions, that achieve the preconditions of the public actions, or private goals.

If a plan that was computed over the determinized CPPP problem fails, instead of replanning, the *chosen* agent that experiences the failure first attempts to return to the plan. The agent does so by replanning independently on a single-agent determinization of its own actions only, with the goal to achieve the preconditions of its next public action.

This approach allows us to avoid the costly joint planning episode, requiring only a single agent to replan, typically for a much shorter horizon. If the agent manages to identify such a plan, it continues to execute this repaired plan until the public action, and then returns to the original plan. If not, then a joint effort to create a new plan is launched.

## Empirical Evaluation

We now provide an empirical evaluation of our methods on domains adapted from the CPPP literature. The implementation is in C#. Experiments were run on a Windows machine with an i7-8550U CPU and 8GB of RAM.

## Domains

The domains were taken from the 2015 CODMAP competition (Štolba, Komenda, and Kovacs 2015)[1] adding stochastic effects with varying probabilities. We now provide a brief description of each domain.

**Blocks (BL):** Blocks world problems where blocks must be moved from an original configuration to some goal configuration. Each agent controls an arm that can move a block. We added the following stochastic effects to this domain: Picking up a block succeeds with probability 0.8, otherwise the block remains where it was. Putting down a block on the table always (probability 1.0) succeeds. Stacking a block on another block succeeds with probability 0.8. Otherwise the block falls on the table. Unstacking a block from other block succeeds with probability 0.9. Otherwise nothing changes.

**Colored-Blocks (CBL):** Hefner, Shani, and Stern (2022) created a new version of the blocks domain where the blocks are colored and each agent can move only specific block colors. In addition, holding a block is a private information known only to the agent who holds it. This domain enables the agents to pick up a stack of two blocks and put it down on the table or on another block. Most actions have identical probabilities to the BL instances. Picking up a stack of two blocks succeeds with probability 0.7. With probability 0.2 the stack falls and the blocks fall on the table. With probability 0.1 the action fails and the stack remains where it was. Putting down a stack of two blocks on the table - with probability 0.7 succeeds, with probability 0.3 fails and both block are on the table. Stacking a stack of two blocks on another block succeeds with probability 0.6. With probability 0.4 the action fails and both blocks fall on the table. Unstacking succeeds with probability 0.6. With probability 0.4 the action fails and all blocks fall on the table.

**Colored-Blocks Extended (CBLE):** A new version of the colored-blocks domain with additional side effects. In this domain, when an agent picks up, puts down, or stacks a block from another block, the other block can fall on the table too, accidentally. The actions have identical probabilities to the CBL instances. If an action fails and a block or stack fall on the table, the other blocks also fall on the table.

[1] http://agents.fel.cvut.cz/codmap/

**Depot (DT):** In the depot domain agents are trucks and distributors. Distributors have hoists capable of lifting and loading or unloading crates onto or off trucks. Trucks move crates between locations. We add the following stochastic effects: Lifting or loading a crates succeeds with probability 0.8. Otherwise the crates remains where it was. Driving between locations succeeds with probability 0.8 succeeds. Otherwise the truck remains where it was.

**Driverlog (DL):** In the driverlog domain there are drivers who can walk between trucks, and drive trucks between locations via paths. There are different paths used for walking or driving. Drivers can board or disembark from a truck and the trucks can be loaded or unloaded with packages. The goal is to bring packages and trucks to target locations. For each action in the domain we had set probability 0.8 for success. If an action failes the state does not change.

**Driverlog Extended (DLE):** A new version of the driverlog domain with additional side effects. For load and unload actions, the action succeeds with probability 0.7. With probability 0.2 the action fails, and the driver disembarks from the truck. Otherwise, the state does not change.

**Elevators (EL):** The elevators domain describes elevators moving between building floors. There are passengers located in initial floors and the goal is to move passengers to target floors. Each action in the domain succeeds with probability 0.8 and otherwise the state does not change.

**Elevators Extended (ELE):** This version of elevators was designed to challenge replanning approaches. Here, elevators may malfunction, requiring repair, which may require several actions to succeed. Fast elevators have a higher probability of a malfunction.

**Logistics (LG):** In this domain there are airplanes and trucks which can move packages between airports and cities, respectively. Each action in the domain succeeds with probability 0.8 and otherwise the state does not change.

**Rovers (RV):** In this domain there are rovers that can move between locations. Each rover has a subset of the following abilities: sampling soil, sampling rock, or taking images in several modes. Soil or Rock samples can be found in various locations and image objectives can be visible from various locations as well. The goal is for the rovers to communicate sampled/image data to landers which have to be visible from the location of data communication. Each action in the domain succeeds with probability 0.8 and otherwise the state does not change.

**Rovers Extended (RVE):** This version of rovers was designed to challenge determinization approaches. Here, different rovers have different probabilities for success on a task. Thus, it is beneficial to assign tasks given these probabilities. Task failures are identified only after the rover transmits the results, requiring traveling back to the task location.

**Zenotravel (ZT):** In this domain there are passengers which can be embark or disembark from aircrafts. The aircraft can fly between locations at alternative velocities. For each action in the domain we set a probability of 0.8 for success and 0.2 for failure where the state does not change.

| Domain | | # Solved instances | | |
|---|---|---|---|---|
| Name | # instances | DP | DPR | CTG |
| Simple domains | | | | |
| BL | 20 | 14 | 14 | 12 |
| DT | 25 | 14 | 14 | 9 |
| DL | 20 | 14 | 14 | 15 |
| EL | 20 | 16 | 17 | 14 |
| LG | 31 | 31 | 31 | 28 |
| RV | 12 | 11 | 11 | 10 |
| ZT | 10 | 10 | 10 | 10 |
| Complex domains | | | | |
| CBL | 5 | 5 | 5 | 5 |
| CBLE | 5 | 4 | 4 | 5 |
| DLE | 20 | 14 | 14 | 14 |
| ELE | 20 | 16 | 14 | 12 |
| RVE | 19 | 19 | 19 | 2 |

Table 2: Coverage: amount of instances for each domain and how many were solved by each method.

## Procedure

We compare our 3 approaches for Online-SCPPP (Section 18): selecting actions following a cost-to-go heuristic estimate (denoted CTG below), the deterministic replanning approach (denoted DR) and the deterministic plan repair (denoted DPR). In addition, we compare our methods to the offline RTDP based approaches (Hefner, Shani, and Stern 2022). We reimplemented the PS-RTDP method of (Hefner, Shani, and Stern 2022) (denoted DRTDP), and also 2 variations in the heuristic used for forward search in RTDP, using the same heuristics used in the online solver (denoted DRTDP-CTG, and DRTDP-DP).

For online algorithms, we terminate once the goal was reached, or after 15 minutes. For offline algorithms, we run RTDP until the policy converges within $\epsilon = 0.7$, or 30 minutes have passed (excluding policy evaluation time). We then estimate the average accumulated cost over 50 policy executions. During planning, policy evaluation is performed every 10 trajectories. The cycle-detection sensitivity of the message passing mechanism was identical for all the problems we ran and was set to 4 cycles.

## Results

Table 1 compares offline and online results for two, relatively difficult benchmarks, colored blocks world (CBL), and Driverlog with driver exit effects (DLE). We compare costs, using unit cost, and wall clock runtime. For offline algorithms we compute the runtime until convergence, and for online approaches, we compute the average trajectory runtime. Clearly, this is not a fair comparison, because offline policy evaluation is done once, and policy execution has almost no cost. Still, this allows us to estimate scaling up.

As we can see, offline planners scale poorly, as their runtime grows exponentially even for very small problems. As we show later, the online algorithms scale to much larger problem sizes. For policy quality, we can see that the online algorithms produce comparable costs in some cases. In CBL-3, on the other hand, the offline approach produced

| Domain | | | Avg cost | | | Avg decision time (sec) | | |
|---|---|---|---|---|---|---|---|---|
| | #Agents | #Objects | DP | DPR | CTG | DP | DPR | CTG |
| BL-1 | 4 | 9 | **29.7 ± 3.24** | 31.1 ± 4.85 | 1590.45 ± 584.93 | 0.25 ± 0.1 | 0.13 ± 0.04 | 0.05 ± 0.01 |
| BL-8 | 4 | 12 | **37.35 ± 3.9** | 38.15 ± 5.58 | - | 0.3 ± 0.12 | 0.14 ± 0.03 | - |
| BL-9 | 4 | 12 | 35.3 ± 3.94 | **34.25 ± 3.43** | 1471.85 ± 1080.23 | 0.34 ± 0.11 | 0.14 ± 0.04 | 0.15 ± 0.05 |
| BL-10 | 4 | 14 | 38 ± 5.02 | **37.05 ± 4.55** | - | 0.35 ± 0.13 | 0.18 ± 0.04 | - |
| DT-1 | 3 | 5 | **11.5 ± 1.24** | 11.75 ± 1.04 | 13.6 ± 2.03 | 0.12 ± 0.02 | 0.12 ± 0.02 | 0 ± 0 |
| DT-6 | 5 | 4 | **11.25 ± 1.7** | 11.35 ± 1.35 | 11.45 ± 1.36 | 0.13 ± 0.03 | 0.12 ± 0.03 | 0.02 ± 0 |
| DT-8 | 5 | 11 | **42.05 ± 2.82** | 44.3 ± 2.57 | 1163.25 ± 696.74 | 0.1 ± 0.01 | 0.09 ± 0 | 0.01 ± 0.01 |
| DT-9 | 8 | 18 | **29.1 ± 1.61** | **29.1 ± 2** | 40.65 ± 7.88 | 0.15 ± 0.02 | 0.15 ± 0.01 | 0.31 ± 0.04 |
| DL-1 | 2 | 4 | 7.65 ± 1.06 | 7.85 ± 1.11 | **2.35 ± 0.57** | 0.25 ± 0.04 | 0.25 ± 0.05 | 0.02 ± 0.01 |
| DL-6 | 3 | 8 | 10.35 ± 1.68 | 10.1 ± 1.41 | **8.1 ± 2.12** | 0.26 ± 0.04 | 0.26 ± 0.04 | 0.07 ± 0.01 |
| DL-9 | 2 | 9 | **27.5 ± 2.54** | 28.05 ± 1.94 | 31.6 ± 7.3 | 0.09 ± 0.01 | 0.08 ± 0.01 | 0.09 ± 0.01 |
| DL-13 | 2 | 9 | **41.45 ± 2.91** | 43.45 ± 3.29 | 96.6 ± 58.24 | 0.11 ± 0.01 | 0.1 ± 0.01 | 0.61 ± 0.11 |
| DL-14 | 3 | 9 | 49.6 ± 3.5 | **48.9 ± 3.51** | 120.85 ± 124.86 | 0.08 ± 0.01 | 0.08 ± 0.01 | 0.61 ± 0.14 |
| EL-1 | 4 | 4 | **25.65 ± 2.78** | 25.95 ± 2.85 | 27.55 ± 8.73 | 0.14 ± 0.02 | 0.13 ± 0.02 | 0.23 ± 0.03 |
| EL-6 | 4 | 9 | 69.05 ± 4.9 | **68.3 ± 4.21** | 312.25 ± 316.83 | 0.06 ± 0.01 | 0.06 ± 0 | 0.34 ± 0.07 |
| EL-9 | 4 | 12 | **73.25 ± 4.21** | 73.5 ± 4.33 | 83.95 ± 36.08 | 0.07 ± 0 | 0.07 ± 0 | 0.93 ± 0.15 |
| EL-13 | 4 | 12 | 106.15 ± 4.49 | 108.05 ± 4.42 | **100.6 ± 32.52** | 0.1 ± 0.01 | 0.1 ± 0.01 | 3.39 ± 0.52 |
| EL-14 | 4 | 14 | 178 ± 5.34 | **177.5 ± 5.85** | - | 0.09 ± 0.01 | 0.09 ± 0 | - |
| LG-1 | 3 | 1 | 12.8 ± 1.29 | 13.05 ± 1.56 | **11.7 ± 1.19** | 0.15 ± 0.02 | 0.15 ± 0.02 | 0 ± 0 |
| LG-10 | 4 | 3 | 23.55 ± 1.86 | 23.65 ± 1.35 | **17.3 ± 1.19** | 0.1 ± 0.01 | 0.1 ± 0.01 | 0.01 ± 0 |
| LG-20 | 5 | 12 | 70.6 ± 3.25 | **69.4 ± 3.58** | 135.8 ± 55.39 | 0.05 ± 0 | 0.05 ± 0 | 0.12 ± 0.02 |
| LG-27 | 7 | 15 | **129.75 ± 4.36** | 130.1 ± 3.88 | 189.2 ± 38.77 | 0.05 ± 0 | 0.05 ± 0 | 0.48 ± 0.05 |
| LG-28 | 7 | 15 | 116.5 ± 6.64 | 114.45 ± 4.95 | **110.5 ± 26.38** | 0.04 ± 0 | 0.05 ± 0 | 0.61 ± 0.09 |
| RV-1 | 2 | 3 | 18.1 ± 1.97 | 18.4 ± 1.88 | **7.25 ± 1.34** | 0.12 ± 0.03 | 0.2 ± 0.18 | 0.01 ± 0 |
| RV-2 | 4 | 6 | 31.4 ± 2.94 | **31.15 ± 2.69** | 179.4 ± 145.05 | 0.11 ± 0.06 | 0.19 ± 0.12 | 0.11 ± 0.06 |
| RV-3 | 4 | 11 | 54.85 ± 3.66 | 55.25 ± 3.63 | **47.4 ± 15.01** | 0.08 ± 0.04 | 0.1 ± 0.05 | 0.36 ± 0.1 |
| RV-4 | 4 | 10 | **82.25 ± 4** | 85.3 ± 5.12 | - | 0.04 ± 0.02 | 0.05 ± 0.03 | - |
| RV-5 | 4 | 11 | 69.9 ± 4.09 | **68.8 ± 2.23** | 615.85 ± 461.9 | 0.06 ± 0.02 | 0.06 ± 0.03 | 0.1 ± 0.05 |
| ZT-1 | 2 | 4 | 8.95 ± 1.5 | 9.75 ± 1.97 | **6.6 ± 1.16** | 0.22 ± 0.03 | 0.21 ± 0.04 | 0.08 ± 0.01 |
| ZT-3 | 2 | 4 | 22 ± 2.14 | 22.85 ± 2.08 | **9.95 ± 1.5** | 0.1 ± 0.02 | 0.09 ± 0.01 | 0.12 ± 0.05 |
| ZT-9 | 3 | 8 | 42.15 ± 2.82 | 41.65 ± 2.85 | **26.35 ± 5.35** | 0.07 ± 0.01 | 0.08 ± 0.01 | 1.22 ± 0.2 |
| ZT-10 | 3 | 10 | 63.6 ± 3.4 | **63.05 ± 3.63** | 74.35 ± 54.15 | 0.06 ± 0.01 | 0.05 ± 0 | 0.8 ± 0.24 |

Table 3: Simple domains: cost and decision time, reporting means and standard error. Best costs are bolded.

substantially better policies. This is expected, as online approaches typically do not have optimality guarantees.

From the offline approaches, the $h_{FF}$ heuristic produced the best results, both in runtime and in policy quality. In the larger domains, the offline approaches did not manage to run a sufficient number of RTDP trajectories in the given 30 minutes, and the resulting policy was of very poor quality.

We now move to analyzing our online methods. Table 2 summarizes the number of instances that we created for each problem (based on the CODMAP problem instances), and how many were solved by each method. For each method and problem we run 20 trials, and report averages over these trials. If at least one of the trials did not reach the goal within the 15 minutes timeout, then we say that the method did not manage to solve the problem. In 5 domains DP and DPR managed to solve larger instances than CTG. Many domains leave much room for improvement in future research.

Tables 3 and 4 show a few examples from each domain (due to the lack of space), comparing the online approaches in terms of average cost and average time for each action decision in seconds. The domains are based on the CODMAP benchmarks (Štolba, Komenda, and Kovacs

2015), with stochastic effects, as specified above. In most domains, we manage to solve many problems, but in some, only a handful of problems are solved.

The CTG heuristic provides the worst results in most problems, except for Logistics (LG) and ZenoTravel (ZT). In the domains where it fails, while it provides very rapid decisions compared to the other methods, it often makes bad action choices, leading to very long execution sequences in many problems. The variations in performance are also very large for the CTG method, where in some cases the method results in very long trajectories. CTG succeeds in LG and ZT, because in these domains the relaxed plan that it computes does not contain conflicting actions. Hence, in these domains the relaxed plan is directly applicable in the original problem. In domains where it succeeds, CTG requires more time than DP and DPR, due to the large branching factor, forcing us to compute the heuristic for many states.

The DP method, which replans every time an unexpected result has was observed, and the DPR method, which uses local single agent replanning, produce very similar plan quality. In many domains, the repair time is considerably lower than the complete replanning time. This is most pro-

| Domain | | | Avg cost | | | Avg decision time (sec) | | |
|---|---|---|---|---|---|---|---|---|
| | #Agents | #Objects | DP | DPR | CTG | DP | DPR | CTG |
| CBL-1 | 2 | 2 | **4.45 ± 0.74** | 4.85 ± 1.35 | **4.45 ± 0.67** | 0.46 ± 0.09 | 0.42 ± 0.1 | 0.01 ± 0.02 |
| CBL-3 | 3 | 4 | 10.75 ± 3.11 | **9.8 ± 1.75** | 13.55 ± 9.81 | 0.25 ± 0.11 | 0.2 ± 0.07 | 0.01 ± 0 |
| CBL-4 | 2 | 5 | **11.75 ± 2.12** | 14.3 ± 3.24 | 22.3 ± 27.03 | 0.19 ± 0.1 | 0.19 ± 0.06 | 0.01 ± 0 |
| CBL-5 | 2 | 6 | **18.5 ± 5.12** | 32.25 ± 8.38 | 120.05 ± 95.05 | 0.23 ± 0.08 | 0.17 ± 0.05 | 0.02 ± 0.01 |
| CBLE-1 | 2 | 2 | 4.45 ± 1.2 | **3.75 ± 0.77** | 3.95 ± 0.97 | 0.47 ± 0.12 | 0.5 ± 0.1 | 0.01 ± 0.02 |
| CBLE-2 | 3 | 3 | 7 ± 1.97 | **6.95 ± 2.36** | 7.65 ± 2.15 | 0.58 ± 0.18 | 0.76 ± 0.35 | 0.01 ± 0 |
| CBLE-3 | 3 | 4 | **9.7 ± 1.98** | 9.75 ± 1.79 | 18.85 ± 10.48 | 0.41 ± 0.16 | 0.51 ± 0.23 | 0.02 ± 0.01 |
| CBLE-4 | 2 | 5 | **12.75 ± 2.19** | 12.8 ± 2.42 | 62.15 ± 63.42 | 0.53 ± 0.16 | 0.58 ± 0.24 | 0.11 ± 0.06 |
| CBLE-5 | 2 | 5 | **15.3 ± 2.7** | 15.65 ± 3.38 | 112.1 ± 64.46 | 0.43 ± 0.13 | 0.57 ± 0.16 | 0.07 ± 0.03 |
| DLE-1 | 2 | 4 | 7.55 ± 1.4 | 7.3 ± 1.14 | **2.4 ± 0.66** | 0.26 ± 0.04 | 0.27 ± 0.05 | 0.02 ± 0.01 |
| DLE-6 | 3 | 8 | 12.05 ± 3.32 | **10.4 ± 2.13** | 10.7 ± 3.49 | 0.28 ± 0.08 | 0.3 ± 0.08 | 0.08 ± 0.01 |
| DLE-10 | 2 | 9 | **22.9 ± 2.91** | 23.8 ± 5.39 | 40.85 ± 8.19 | 0.3 ± 0.09 | 0.25 ± 0.05 | 0.19 ± 0.02 |
| DLE-13 | 2 | 9 | **38.75 ± 6.46** | 44.15 ± 3.66 | 275.3 ± 94.91 | 0.27 ± 0.1 | 0.18 ± 0.03 | 0.53 ± 0.11 |
| DLE-14 | 3 | 9 | 54.95 ± 6.23 | **53.2 ± 4.25** | 341.3 ± 260.99 | 0.25 ± 0.13 | 0.14 ± 0.04 | 0.59 ± 0.12 |
| ELE-3 | 4 | 6 | **47.1 ± 10.42** | 49.35 ± 11.11 | 49.55 ± 12.73 | 0.25 ± 0.05 | 0.49 ± 0.3 | 0.27 ± 0.02 |
| ELE-4 | 4 | 7 | **70.05 ± 11.94** | 71 ± 10.32 | 226.85 ± 162.9 | 0.27 ± 0.04 | 0.4 ± 0.1 | 0.26 ± 0.05 |
| ELE-5 | 4 | 8 | **78.75 ± 13.06** | 82.95 ± 11.36 | 79 ± 51.45 | 0.31 ± 0.07 | 0.57 ± 0.26 | 0.44 ± 0.07 |
| ELE-6 | 4 | 9 | 93.8 ± 13.13 | 104.2 ± 14.97 | **91.55 ± 21.63** | 0.31 ± 0.08 | 0.39 ± 0.1 | 0.49 ± 0.06 |
| ELE-8 | 4 | 11 | 86.6 ± 11.28 | 96.65 ± 13.67 | **75.75 ± 34.2** | 0.31 ± 0.04 | 0.51 ± 0.17 | 0.84 ± 0.14 |
| ELE-9 | 4 | 12 | 111.75 ± 13.97 | 110.15 ± 11.46 | **102.25 ± 29.52** | 0.34 ± 0.08 | 0.56 ± 0.23 | 0.79 ± 0.12 |
| ELE-10 | 4 | 13 | **135.9 ± 16.89** | 138.75 ± 14.13 | 162.65 ± 51.05 | 0.35 ± 0.04 | 0.58 ± 0.1 | 1.04 ± 0.16 |
| ELE-11 | 4 | 8 | 96.3 ± 7.33 | 96.3 ± 11.89 | **81.65 ± 40.12** | 0.41 ± 0.06 | 1.1 ± 0.96 | 1.17 ± 0.19 |
| ELE-13 | 4 | 12 | 159.4 ± 17.18 | 155.75 ± 22.23 | **109.7 ± 24.97** | 0.69 ± 0.08 | 2.21 ± 0.96 | 2.29 ± 0.18 |
| RVE-2 | 4 | 9 | **59.15 ± 9.59** | 67.75 ± 13.86 | - | 0.37 ± 0.13 | 0.27 ± 0.06 | - |
| RVE-3 | 4 | 6 | 39.75 ± 12.72 | **37.4 ± 10.47** | 120.7 ± 23.94 | 0.06 ± 0.02 | 0.05 ± 0.01 | 0.29 ± 0.19 |
| RVE-4 | 4 | 12 | **79.55 ± 9.51** | 79.95 ± 10.46 | 579 ± 84.6 | 0.05 ± 0.02 | 0.05 ± 0.08 | 0.32 ± 0.01 |
| RVE-5 | 4 | 8 | 68.1 ± 17.11 | **59.3 ± 12.61** | - | 0.17 ± 0.04 | 0.12 ± 0.02 | - |
| RVE-9 | 6 | 11 | **68.2 ± 7.05** | 70.8 ± 9.73 | - | 0.2 ± 0.19 | 0.08 ± 0.01 | - |
| RVE-11 | 8 | 20 | 218.25 ± 19.31 | **215.15 ± 22.58** | - | 0.26 ± 0.09 | 0.1 ± 0.01 | - |

Table 4: Complex domains: cost and decision time, reporting means and standard error. Best costs are bolded.

nounced in the larger instances of the blocks domain (BL), and the larger instances of the driverlog problem (DLE). In CBLE, where different agents move different blocks, and one agent can knock down blocks of other agents, plan repair often fails, resulting in longer decision time, as a complete replan is needed after the local replan fails.

In the extended elevator domain (ELE), the determinization ignores malfunctions, and assumes that repairs always succeed, thus insisting on using the faster (malfunctioning) elevators. Both DP and DPR provide worse plans that CFG, which takes success probabilities into consideration. On the other hand, CTG is slower, and hence, does not scale as well.

In extended rovers problems, where one should assign tasks to rovers that are most likely to succeed, the DP and DPR methods manage to solve the problems rapidly, but do not assign tasks optimally, as the determinization ignores the probabilities. This is not reflected in the CTG method, because the particular heuristic that we constructed, although taking probabilities into account, fails to provide good estimates, and thus, does not provide good solutions, and fails utterly in larger problems.

The results above show that simple heuristics, based on an estimation of the cost-to-go are less effective for online SCPPP, while replanning approaches work well. Local plan repairs can be effective, reducing the average decision time

considerably in some cases.

## Conclusion

In this paper we suggested an online planning approach for stochastic multi agent planning problems under privacy constraints. We suggest heuristics for deciding online on the next action to execute. We show that replanning-based heuristics are much more effective than a cost-to-go estimate, and that local plan repairs can be used to reduce the decision time in some cases. We show that, while compromising on optimality guarantees, our online approach scale to much larger problems than offline approaches, vastly extending the range of problems that can be approached.

There are many future directions for online methods for stochastic CPPP. First, it is important to construct additional benchmarks that present interesting stochastic properties. One can draw inspiration from problems suggest for single agent stochastic planning competitions in the past, as opposed to the CODMAP benchmarks that we build upon. It is also interesting to explore additional online algorithms, most importantly, UCT-based approaches. It is also interesting to explore additional heuristics that were constructed for CPPP problems, such as the dependency projection (Maliah, Shani, and Stern 2016).

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
