# OpenReview forum: "Online Planning for Stochastic Collaborative Privacy Preserving Planning"
_icaps-conference.org/ICAPS/2023/Workshop/HSDIP — ICAPS HSDIP 2023_

### Official Review · Reviewer_ieNk · 2023-04-26

**Rating:** 7
**Confidence:** 4

**Review:**

The paper presents new online approaches to solve stochastic collaborative
multi-agent privacy preserving planning (SCPPP). In this type of planning,
there is a set of public variables, and for each agent a set of private
variables. Furthermore, each agent has a set of private and a set of public
actions. All agents work together on achieving the goal.
All three approaches work by iteratively computing which agent gets to apply
the next action, and this agent then selecting its best action and sharing the
public part of successor state with all other agents. The first approach CTG
(cost-to-go) computes a stochastic and privacy preserving version of the FF
heuristic, which in its relaxation considers all possible effects of a
stochastic action as achieved. The second approach DR computes an overall plan
for the determinization of the problem and replans whenever the execution of
this plan fails. The third approach DPR also starts with a plan for the
determinization, but attempts to locally repair the plan if it fails before
replanning.
Experimental results show that the proposed approaches manage to solve
significantly larger problems than existing offline ones, while somewhat
suffering in solution cost as expected. Between the three new approaches, CTG
decides fast but can produce substantially worse plans. DR and DPR perform
similar, but DPR is sometimes faster if the local plan repair succeeds.

The paper is thematically a good fit for HSDIP and presents interesting
results. It builds on existing approaches for both stochastic planning and
collaborative multi-agent privacy-preserving planning, but their combination
for SCPPP is novel. The paper is generally well structured and easy to follow,
and I in particular liked how the experimental evaluation explains why certain
approaches work good/bad on certain domains.

I did however notice some minor clarity issues:
1) The formal definition of SCPPP is not detailed enough, mainly the definition
of private actions. I assume that private actions are only over private
variables, but this is never stated.
2) I don't understand how the distributed computation of FF works. You first
talk about a delete relaxation state space, but then about constructing a
state. Do you mean the later mentioned fact graph here? Furthermore, you say
that opposed to normal FF, a fact can be achieved multiple times in different
layers and later achievements might have lower cost. I don't understand why
this is the case. Is the probability of a fact happening somehow integrated in
the cost? Finally, I don't really understand how the distributed computation
works. I imagine some form of iterative process where all agents first compute
the fact graph for only their actions, then all graphs are somehow merged and
the agents recompute what they now can newly apply on this merged graph.
3) Tables 3 and 4 are very hard to parse. I think plots would work much better
here to get an intuitive overview how often and where the numbers differ
significantly. Maybe 3 scatterplots (CTG-DR, CTG-DPR, DR-DPR), where you plot
all 20 trials and use transparency? Or a histogram with the x axis being the
selected problems?
4) In the subsection Procedure, the last paragraph talks about performing
policy-evaluation every 10 trajectories and cycle detection. I did not
understand what trajectories and cycle-detection mean here.

Overall, I recommend to accept the paper, it presents an interesting
combination of existing techniques for the new SCPPP problem, and the
evaluation not only shows that the approach is competitive but also gives some
insight on which properties of a given domain are relevant for certain
approaches to be successful.

Minor comments:
 - last paragraph of Introduction: "We first shows" -> show
 - Background, definition of plan: I would not use k in $\pi=(a_1, \dots , a_k)$
 since k is already used to denote the number of agents.
 - Algorithm 1: you use variables $s_c^i$ before initializing it. I also wonder
 what exactly it is. I would assume it is actually the tuple $<s_\text{pub}, s_1^\sim, \dots,
 s_k^\sim>$, but why is it then called $s_c^i$?
 - Algorithm 1: How does it terminate? Wouldn't you need to check somewhere if
 all agents report goal?
 - Table 2-4: Is there a reason to change the order such that CTG is last? Up
 until this point CTG was always the first approach considered, and the change
 threw me off.
 - Table 2: I would put the best results in bold as with other tables.
 - From Table 2 onward you write DP instead of DR
 - Results, paragraph about Tables 3 and 4: You say in some domains only a
 handful of problems are solved, but in Table 2 it looks like all domains are
 solved fairly well except mabye DT.
 - Results, second last paragraph on page 7: You say that a large branching
 factor is problematic for CTG, but is this not also the case for DR and DPR?
 After all it also performs heuristic search, and I thought the branching
 factor is the same for the determinization.
 - Results, last paragraph of page 7: "an unexpected result has was observed"
 -> an unexpected result was observed
 - Conclusion: Two consecutive sentences start with "It is also interesting"
 - References: Missing page number for Brafman 2015, Keller and Eyerich 2012,
 Maliah et al 2016, Stolba and Komenda 2014
 - References are very inconsistent in whether they use abbreviations for
 conferences/journals or not

Questions to the authors:
1) Can you elaborate on the computation for FF? (see above)
2) CPPP somewhat reminded me of decoupled search, where the variables are split
to "center" variables and sets of "leaf variables" which only interact with
global variables but not with other leafs. However, in decoupled search there is no
privacy concerns, so am I right in assuming that decoupled search cannot be
used in this context?
3) When an agent locally replans in DPR, what does it plan towards? To its
goal, the overall goal, or to some state of the original plan?

---

### Official Review · Reviewer_dFXH · 2023-04-26
**Great addition to the workshop**

**Rating:** 7
**Confidence:** 4

**Review:**

The paper presents approaches to tackle stochastic multi-agent planning problems in a privacy-preserving setting. Here, a set of agents has to collaborate to achieve a set of goals, which can be joint or individual to an agent. The privacy-preserving aspect is that every agent has a set of private facts and actions that the others don't know about. The public part of such a planning task, facts and actions, is known to every agent and used as a basis to synchronize.
A key aspect of the presented work is that the planning tasks are not solved optimally, as is usually done in stochastic settings to obtain a full policy, but the agents take decisions online. As a result, the resulting solutions are very likely sub-optimal, and the process can even get stuck and will have to be restarted. On the positive side, the online approach is a lot more efficient and can solve tasks that are significantly larger than those solvable using an offline policy generation.
The paper builds on existing algorithms such as FF-replan, which is known from stochastic single-agent planning, and PS-RTDP, which is based on the RTDP algorithm that solves general MDP planning problems. Concretely, the algorithm coordinates the agents by sending messages from each agent to the others if one of them reaches its private goal state, or a novel public state. On a high level, this works in two phases: 1) given the current public state, the agent that is going to take the next action is selected, and then 2) this agent makes steps until reaching a new public state.

The authors propose three methods to take the involved decisions:
1) A variant of the hFF heuristic, which was previously adapted to privacy-perserving multi-agent planning, is used to decide which agent is most the best choice in the current state, as well as which action that agent takes. To do so, the heuristic is evaluated multiple times, for each agent, and to decide on the best action of that agent.
2) An approach based on FF-replan that works on a determinization of the stochastic problem and replans if unexpected action outcomes occur.
3) A variant of 2) that does not globally replan when an unexpected action effect occurs, but only locally for the agent that experiences that effect.

The authors evaluate their approach on a set of existing planning tasks that have been made stochastic. The results show that their approach works very well and has favorable scaling behavior compared to offline algorithms.

The paper is a great fit for the workshop. The topic is clearly relevant, the presentation is clear, and the shown approach give good results.

Question:
Is your approach complete in case where "parallel execution" is required, i.e. if more than one agent has to perform an action before obtaining the next public state?

Minor things:
- last paragraph in intro: "We first showS"
- bottom right of page 2: "in THE state space"
- bottom left of page 3: "at by"
- one paragraph later: "execution OF its applicable actions"
- experimental setup: did you specify a runtime limit? are planners executed single-threaded?
- bottom right of page 6: "both blockS"
- caption of Table 2: "NUMBER of instances"

---

### Decision · Program_Chairs · 2023-05-05

**Decision:**

Accept

**Comment:**

We are happy to announce that the paper is accepted to the workshop. Both reviewers clearly stated that the work is very relevant for the workshop.

For the final version, we ask you to address the comments made by the reviewers. It would also be good if you could incorporate clarifications to the questions that were asked.